# Accelerometer-Measured Physical Activity and Sedentary Behavior Levels and Patterns in Female Sixth Graders: The CReActivity Project

**DOI:** 10.3390/ijerph18010032

**Published:** 2020-12-23

**Authors:** Joachim Bachner, David J. Sturm, Yolanda Demetriou

**Affiliations:** Department of Sport and Health Sciences, Technical University of Munich, 80992 Munich, Germany; david.sturm@tum.de (D.J.S.); yolanda.demetriou@tum.de (Y.D.)

**Keywords:** accelerometry, secondary school, female, sedentariness, activity

## Abstract

Regular physical activity (PA) and low levels of sedentary behavior (SB) have positive health effects on young people. Adolescent girls of low socioeconomic background represent a high-risk group with regard to physical inactivity and SB. In this study, accelerometer-measured levels of PA and SB of female sixth graders attending lower secondary schools in Germany are presented, patterns of PA and SB throughout the day are described and differences between weekdays and weekend days are analyzed. Data of 425 students of the CReActivity project were analyzed. Sampling and processing of accelerometer data followed recent recommendations, which had not been applied to data of a German-speaking sample before. The WHO recommendation of daily 60 min moderate-to-vigorous PA was fulfilled by 90.4% of the girls on weekdays and by 57.4% on weekend days. The significant weekday–weekend differences were mainly associated with active commuting to and from school. Students engaged in SB for more than 8 h on weekdays and for over 7 h on weekend days. The results suggest a strong need for interventions increasing PA and reducing SB, especially during school hours and on weekends. Furthermore, a comparison with methods and results of previous studies highlights the need to follow recent criteria in accelerometer data sampling and processing to ensure an accurate and valid differentiation between PA-related risk groups and non-risk groups.

## 1. Introduction

The World Health Organization (WHO) recommends that children and adolescents aged 5 to 17 years engage in moderate-to-vigorous intensity physical activity (MVPA) for an average of 60 min per day [1]. In Germany, the recommendations are even higher, emphasizing that children and youth should accomplish at least 90 min of daily MVPA [2]. Recent systematic reviews suggest that children who follow these recommendations are at a lower risk of overweight or obesity, type II diabetes mellitus and metabolic syndrome and have higher fitness levels than children who do not meet the MVPA recommendations [3,4].

When recommendations on physical activity (PA) are not fulfilled, the person’s behavior is classified as physical inactivity. However, physical inactivity is not the same as sedentary behavior (SB). Instead, SB specifically comprises any waking behavior that is characterized by an energy expenditure of ≤1.5 metabolic equivalents (METs) while being in a sitting, reclining or lying posture (e.g., use of electronic devices while sitting, reclining or lying; sitting at school or in public transport) [5]. Sedentary time (ST) indicates the time spent in sedentary behaviors. The SB pattern finally indicates how a person accumulates SB throughout a given time period, such as a day or a week, while being awake. These patterns refer to the timing, duration and frequency of sedentary bouts, i.e., time periods of uninterrupted ST and breaks between sedentary bouts [5]. Canadian and American recommendations emphasize that for gaining health benefits, children (aged 5–11 years) and youth (aged 12–17 years) should limit recreational screen time to no more than 2 h per day and limit sedentary motorized transport, extended sitting time and time spent indoors throughout the day [6]. According to the German recommendations, children should spend a maximum of 60 min sedentary per day and adolescents a maximum of 120 min per day during recreational time [2]. A school child can both accumulate ten hours of ST during a school day and fulfill PA recommendations by going for a 60 min run in the evening. In this case, the child would accumulate an unhealthy amount of ST but would not be classified as physically inactive. Thus, physical inactivity and SB represent different aspects of PA behavior, which to some extent imply different risk factors for health. Sufficient MVPA levels may only compensate for the health risks of high ST levels to a certain degree [7]. Higher ST is associated with unfavorable body composition and behavioral conduct, higher clustered cardiometabolic risk, lower fitness and lower self-esteem in boys and girls [8].

PA and SB are complex multifactorial behaviors that are difficult to assess. Hence, to obtain accurate data, device-based measures of PA and SB are recommended. The use of self-report questionnaires to determine ST and MVPA reveals some, but not all, relationships between PA and health risk factors [9]. Triaxial accelerometers, such as the Actigraph GT3X+, are one of few devices of sufficient sensitivity to detect small movements during sitting and standing. Although they are not free of shortcomings, such as inadequate measurement while cycling or swimming or the dependence of PA values on different approaches in collecting and processing accelerometer data (e.g., sampling rate, cut points for different PA intensities), accelerometers have become the preferred choice for assessing different levels of PA in children and adolescents [10].

Internationally, the use of accelerometers to assess PA and SB in children is increasing [11]. However, the 2018 German Report Card on Physical Activity of Children and Youth, which aims to evaluate and benchmark the national PA promotion efforts in children and adolescents, showed that in Germany these studies are sparse [12].

Self-report data from wave two of the population-based German National Health Interview and Examination Survey for Children and Adolescents (KiGGS) revealed that 22.4% of the girls and 29.4% of the boys between 3–17 years met the WHO recommendations [13]. In the German national Health Behaviour in School-aged Children (HBSC) survey (2013/14), self-report data received from 11 to 15-year-olds showed that only 14% of girls and 19.9% of boys were physically active at moderate-to-vigorous intensity for at least 60 min per day [14]. So far, only three studies provided PA data of larger samples of children and adolescents with accelerometers. The Identification and prevention of dietary- and lifestyle-induced health effects in children and infants study (IDEFICS) conducted accelerometer measurements of PA of children aged two to ten years (N = 1037, mean age = 6.5 years). The proportion reaching 60 min MVPA on a daily basis was 14.0% for girls and 33.3% for boys [15]. The German Infant Nutrition Intervention Programme study (GINIplus) asked potential participants at the age of 15 to take part in accelerometer assessments (N = 269, mean age = 15.5 years, SD = 0.3). The WHO recommendation was fulfilled by 17.2% of the girls and 24.9% of the boys [16]. In wave two of the KiGGS study, PA was assessed with self-report questionnaires (N = 12981) and, in a subsample, with accelerometers (N = 2278). Initial results of accelerometer data analysis showed that 19.7% of the girls and 36.6% of the boys aged 6 to 17 years exhibited 60 min daily of MVPA [17]. These numbers reflect the typically found effects of gender and age [18], indicating that on average, boys and younger children are more active than girls and older adolescents, respectively.

Similarly, only a few studies that assess SB of children and adolescents living in Germany in a device-based manner exist. Self-report data from the KiGGS and HBSC studies in Germany showed that 60.4 to 72.3% of boys and 55.6 to 57.8% of girls spent more than two hours per day watching TV and using electronic media [14,19]. Based on their study, Huber and Köppel [20] concluded that children and adolescents spent 71% of their awake time on weekdays and 54% on weekend days sedentary. Finally, based on accelerometer data from the GINIplus and IDEFICS studies, children and adolescents spent roughly two thirds of their awake time in a sedentary position [15,16].

PA levels and SB for children vary throughout the day as well as between weekdays and weekends. On average, school children are more active on weekdays than on weekend days [21]. Furthermore, on weekdays, peaks of PA can be seen during the commute to school in the morning, during lunch break and directly after the end of school hours [22]. Additionally, PA and SB levels and patterns of children and adolescents vary between countries due to cultural and environmental differences [18].

According to the socioecological framework, health behaviors like PA and SB can be influenced by factors at the intrapersonal level (e.g., gender or psychological constructs), interpersonal level (e.g., group processes), organizational level (e.g., schools), community level (e.g., built environment, local infrastructure) and policy level (e.g., laws). It is further assumed that behavior is affected by direct, indirect and interactive effects of the respective levels [23]. Although the most proximal factors of the intrapersonal level seem to be more influential on PA and SB compared to factors of the social or built environment, responsibility for the population’s health behavior is supposed to be shared between the actors from the different levels of influence [23,24]. In line with this notion, family and school represent the two main settings where interventions to increase PA and reduce SB of school children are conducted [25]. Within the school setting, numerous PA programs have been conducted, including a substantial number of intervention studies that generated positive effects on accelerometer-measured PA. These programs can be further divided into classroom-based interventions [26], interventions during recess [27], interventions focusing on active school transport [28] and after-school interventions [29]. As a first step toward the development of a potentially effective intervention program, it is necessary to differentiate in which situations or settings PA should be promoted in order to appropriately respond to existing deficits and challenges [25]. Therefore, understanding differences in PA and SB of German students during the day and over the course of the week is of importance when planning interventions. Thus, the aim of this contribution is (a) to describe the level of ST, light physical activity (LPA) and MVPA in a sample of female sixth graders in Southern Germany, (b) to report the percentage of students complying with the WHO recommendations of MVPA (60 min per day) and the German recommendations of MVPA (90 min per day) and ST (less than two hours per day), (c) to provide an overview of patterns of PA and ST throughout the day and d) to analyze differences between weekdays and weekends in these values and patterns. The study focuses on female sixth graders of lower secondary schools as they represent a specific risk group for physical inactivity in terms of gender, age and socioeconomic status (SES) [13,18]. It was assumed that the majority of female students did not meet the WHO and German recommendations regarding MVPA and ST. Furthermore, MVPA levels were expected to be higher on weekdays compared to weekend days, including peaks for transport to and from school.

Insights are generated through an analysis following the most recent recommendations of accelerometer-related literature [30], which to date have not been applied to data of a German-speaking sample. Comparisons to existing studies analyzing PA and SB in children and adolescents in Germany, Europe and worldwide, as well as implications of different decisions regarding sampling and processing of accelerometer data, are discussed in detail.

## 2. Materials and Methods

### 2.1. Participants

The original sample comprised 622 sixth graders (77 boys; 12.4%) aged 11 to 12 years from 20 schools. The schools belonged to two kinds of German secondary schools (Realschule and Gymnasium), which represent an intermediate and high secondary education, respectively. Sample recruitment took place both in and around (distance <25 km) the cities of Munich, Tübingen and Freiburg, as there were enough members of the research team regularly at site to guarantee an economic and flexible sampling process with respect to the schedules of school principals, teachers and students. Every student free of any acute injuries (e.g., broken leg) and clinical diagnoses prohibiting PA could participate in the study. Participation rate was 74.1%. The main reasons for not being willing to take part in the study were children’s doubts regarding their outer appearance when wearing the accelerometers and their concerns that the devices might hinder their performance and enjoyment when being physically active.

For the most part, data assessments took place in the context of the CReActivity intervention study focusing on the promotion of girls’ PA [31]. The intervention was implemented in single-sex female physical education (PE) in the sixth grade of lower secondary schools. This school type was chosen because of its higher portion of students with a lower SES compared to other types of secondary schools in Germany [32]. Further details about the intervention program can be found in the study protocol [31]. The data used for the present analysis were taken from pilot studies testing the feasibility of accelerometer assessments (including girls and boys) and from the baseline assessments of the intervention study.

The minimal sample size was calculated under consideration of the intended increase in MVPA provoked by the intervention. Furthermore, using a formula by Rutterford et al. [33], the estimated intracluster correlation, the supposed variation in class sizes and the levels of significance and power were considered as well. Sample size calculation resulted in a minimum of 467 students being required.

The original sample size of 622 students was diminished by failure or loss of assessment devices or by children not wearing the devices for the required amount of time, resulting in invalid measurements. Furthermore, due to their small number, data of boys were excluded to avoid a remarkable imbalance regarding gender that might have led to a bias in the results. Eventually, 425 female sixth graders constituted the sample for the PA data analysis. Average BMI (kg/m^2^) of the final sample was M = 19.5 (SD = 3.7; N = 282). The number of students participating in the measurements for the BMI was reduced by students refusing to get weighed because they were afraid that their classmates might find out about their weight. This concern applied to both apparently normal-weight and overweight girls, although each student was separated from the class for the measurements. SES was assessed by questions concerning the parents’ current jobs. The answers were classified by means of the International Socioeconomic Index of Occupational Status (ISEI) based on the International Standard Classification of Occupation 2008 (ISCO-08) [34]. When both parents were employed, the job with the higher ISEI was considered (HISEI). The HISEI of the final sample ranged between 15 and 89 with a mean value of 50.5 (SD = 16.0; N = 360). HISEI could not be calculated for every participant because of vague answers which made precise classifications impossible.

The dataset analyzed for this study is provided as Appendix A.

### 2.2. Measures

#### Physical Activity

PA was measured with accelerometers (ActiGraph models GT3X to wGT3X-BT; Pensacola, FL, USA). A small part of the sample (46 students) was assessed with the GT3X model, while the vast majority wore newer models. The agreement regarding PA outcomes between the different generations of ActiGraph models used in this study was high, with the differences in MVPA values measured with different models being close to zero [35]. The recent systematic review of Migueles et al. served as the main foundation for choosing the accelerometer data collection and processing criteria [30].

The devices were worn on the right hip with an elastic belt. Hip placement was chosen since it classifies PA behavior at least as accurately as wrist placement, with the wearing compliance being comparable for both placements [36]. The participants had to wear the accelerometers for at least seven consecutive days. On weekdays children put on the devices when they started their way to school at the latest, on weekend days they put them on right after getting up. The accelerometers had to be worn all day long, except during water-based activities, until 9 p.m. or until they went to bed. ActiLife (v. 6.13.3, ActiGraph, Pensacola, FL, USA) was used for initialization of accelerometers and the processing of the assessed data. The sampling rate was set to 30 Hz.

When downloading the data from the accelerometers, all three axes (vertical, horizontal, medio-lateral) were used to calculate the vector magnitude (VM) activity counts, which were summed over 1-s epochs (10-s epochs for the GT3X model because of lower battery and memory capacity), considering the intermittent activity behavior of children [30]. Furthermore, choosing a short epoch length means that the resulting resolution of the measurement is higher [37]. VM was calculated as the square root of the sum of the three axes’ squared activity counts [38]. Data filtering with the ActiLife low frequency extension (LFE) was not applied.

As a next step, wear-time validation of the data was conducted with the algorithm by Choi, Liu, Matthews and Buchowski [39]. Although the Choi et al. algorithm is most popular in adult samples, it was used in this study. It classifies periods of at least 90 min without any activity counts as non-wear time counts. Up to two minutes of temporary activity are tolerated in the classification of a period as non-wear time as long as there are 30-min windows with no activity counts before and after this allowance period [39]. This is a more liberal non-wear time definition, since data are not excluded before there is more than 90 min of zero activity. Instead, these periods are considered as ST. By choosing a longer non-wear time definition, risk groups like adolescents with a higher BMI, who are of special interest when investigating PA, are retained in the analysis to a greater extent. Since they might show longer periods of ST, the risk of misclassifying these periods as non-wear time is lower when more liberal non-wear time definitions are used [30]. Hence, while accepting a decrease in activity counts per minute (CPM), the chosen non-wear time definition prevents the potential loss of data of important risk groups [30]. A participant’s PA data were considered as valid if data were available for at least three weekdays and one weekend day, with at least eight hours of wear time (WT) being required for a valid day. These criteria were seen as a good compromise between obtaining valid PA values and at the same time retaining a comprehensive sample preserving the study’s power [30]. Applying these constraints, 78.0% of the participants included in this study provided valid PA data (425 out of 545 girls).

In a last step, the wear time-validated activity data were analyzed by means of the cut points provided by Hänggi, Phillips and Rowlands to eventually calculate the average duration of ST, LPA and MVPA for the respective periods of interest for each participant [10]. Based on counts per minute (60-s epoch) and counts per second (CPS; 1-s epoch), respectively, the cut points were: less than 180 CPM/3 CPS for ST, 180 to 3360 CPM/3 to 56 CPS for LPA and more than 3360 CPM/56 CPS for MVPA. So for every epoch it is checked whether the respective cut point is exceeded. Then, the epochs in which the cut points are exceeded are added, which finally leads to values normally reported in minutes per day. To obtain accurate values it is important to use cut points which, first of all, offer the most precise assessment and, secondly, were validated using the same criteria in collecting and processing PA data as we used [30]. The cut points from Hänggi et al. were validated with a sample of similar age wearing the devices (GT3X) on the right hip and a 1-s epoch length was used to sum up the VM activity counts without applying the LFE filter. Additionally, raw vector magnitude counts per minute (VMCPM) were given to indicate values which were free from the influence of subsequent transformations or analyses [38].

According to the wearing guidelines described above, PA between 5 a.m. and 9 p.m. was considered for the PA data analysis. In this way, the activities of participants who got up early were included.

### 2.3. Procedure

Before eligible schools were contacted, the study and its associated data assessments were approved by the ethics commission of the Technical University of Munich, registered under 155/16 S, and the Ministry of Cultural Affairs and Education of the state of Bavaria in Germany.

Then, schools of the eligible school forms which were in and around the cities where enough manpower was provided were contacted and could participate voluntarily. After the schools had indicated a general willingness to take part in the study, several weeks before the scheduled beginning of the data assessment, students and their parents were informed in writing about the purpose and the procedure of the assessments. Students did not participate unless they and their parents had provided written consent beforehand. Participation in the assessments was voluntary and neither schools nor students were rewarded in any way. Not taking part did not lead to any consequences for schools or students.

Data were assessed during autumn and winter in 2016, 2017 and 2018. Data assessments started at the beginning of a school lesson. Codes were used to ensure the anonymity of the participants. Before the distribution of the accelerometers, members of the assessment team explained how to put them on. At least 25% of the participants of each class (or more, if there were more volunteers) were handed out a single-sided information sheet explaining how to handle the accelerometers. These students then served as contact persons for the upcoming assessment days in case their classmates had questions, like when and how to wear the devices. Furthermore, the volunteers were to remind their classmates to wear the accelerometers regularly and correctly throughout the assessment days. After one week, the accelerometers were collected in school by a member of the assessment team.

### 2.4. Data Analysis

Statistical data analysis was performed in SPSS, version 25 [40]. PA behavior was described by the means and standard deviations for ST, LPA, MVPA and VMCPM. Additionally, the average wear time for valid days was calculated. Every descriptive statistic was given for an average weekday, weekend day and for an average day considering the respective amount of valid weekdays and weekend days for each participant. Based on the mean values of each participant, the percentage of girls fulfilling the WHO and German recommendations regarding MVPA per day was calculated. To examine patterns of PA and SB throughout the day, the hourly means of valid cases were incorporated into a diagram. The distribution of the ST and PA values at weekdays and weekend days differed slightly from a normal distribution. Although the sample could be considered large enough to still allow the application of parametric tests, differences between the values of weekdays and weekend days were compared with a Wilcoxon test. To be able to test for differences in PA intensity between weekdays and weekend days, the change in percent of WT and MVPA from weekday to weekend was calculated as well as the respective portion of ST, LPA and MVPA during WT. Discernible differences in the ST and MVPA patterns for weekdays and weekend days were further analyzed by first building new variables (e.g., for periods > 1 h). As the distributions in the periods of interest differed significantly from a normal distribution, the differences between weekdays and weekend days were again tested with Wilcoxon tests. Effect sizes r were classified using the criteria from Cohen, which consider r > 0.1 a small effect, r > 0.3 a medium effect and r > 0.5 a large effect [41].

## 3. Results

Table 1 shows the means and standard deviations of ST and PA behavior as well as the average WT for valid days. Figure 1 shows the percentage of the girls that met the WHO and German recommendation for MVPA on weekdays and weekend days, respectively. Whereas 90.4% reached the WHO recommendation of 60 min of MVPA during weekdays, this number diminished to 57.4% for an average weekend day. The German MVPA recommendation of 90 min of MVPA was fulfilled by 42.1% on a weekday and 19.8% on a weekend day. Additionally, on a weekday only 6 of 425 students (1.4%) managed to limit their ST during recreational time (between 4 and 9 pm) to less than two hours. On a weekend day, no one accumulated less than two hours being sedentary.

Table 1 not only indicates that the girls’ MVPA values showed a significant decline from weekdays to weekend days (z = −13.20, *p* < 0.001, r = 0.45), but that ST (z = −14.07, *p* < 0.001, r = 0.48), LPA (z = −13.10, *p* < 0.001, r = 0.45) and the average WT for a valid day (z = −16.21, *p* < 0.001, r = 0.56) also dropped to a similar extent. Furthermore, a comparison of the VMCPM between weekdays and weekend days showed significantly less counts per min on weekend days (z = −2.72, *p* < 0.01, r = 0.09). This raised the question of whether the adolescents’ PA behavior was less intense on weekend days or if this behavior was only performed for a smaller amount of time for the participants getting up later on weekend days compared to weekdays. To answer this question, the extent of the decrease in MVPA and WT from weekdays to weekend days was compared. The decline in percent of MVPA was significantly larger than that of WT (z = −5.62, *p* < 0.001, r = 0.19). Likewise, the portion of MVPA during WT was higher on weekdays compared to weekend days (weekdays = 11.2%, weekend days = 10.5%; z = −5.39, *p* < 0.001, r = 0.18). While there was no difference regarding LPA (weekdays = 22.7%, weekend days = 22.6%; z = −0.63, *p* = 0.53), there were more girls with a higher portion of ST during WT on weekend days than on weekdays (weekdays = 66.0%, weekend days = 66.9%; z = −2.56, *p* < 0.05, r = 0.09).

Figure 2 shows the adolescents’ SB and PA patterns on weekdays and weekends. Between 6 a.m. and 1 p.m., the girls accumulated about 100 min more ST on weekdays than on weekend days (weekdays = 251.78, weekend days = 148.56; z = −16.89, *p* < 0.001, r = 0.58). MVPA was higher on weekdays compared to weekend days, especially in the time periods between 7 a.m. and 8 a.m. (weekdays = 9.53, weekend days = 0.38; z = −17.73, *p* < 0.001, r = 0.61), as well as between 1 p.m. and 2 p.m. (weekdays = 10.11, weekend days = 6.21; z = −12.77, *p* < 0.001, r = 0.44).

Figure 3 displays the PA pattern based on VMCPM. The scale of the *y*-axis ranges from VMCPM representing no activity to the upper limit of LPA defined by the cut points from Hänggi et al. (3360 VMCPM). The main differences in average VMCPM between weekdays and weekends can be observed between 6 a.m. and 8 a.m. (weekdays = 930.28, weekend days = 93.80; z = −17.54, *p* < 0.001, r = 0.60) and between 1 p.m. and 2 p.m. (weekdays = 1201.09, weekend days = 852.74; z = −11.17, *p* < 0.001, r = 0.38).

Whereas SES was not relevant for PA behavior, BMI consistently exhibited significant associations with girls’ PA and SB both on weekdays and weekend days. In total, BMI was positively related to ST (r = 0.24, *p* < 0.001) and negatively related to LPA (r = −0.30, *p* < 0.001), MVPA (r = −0.15, *p* < 0.05) and VMCPM (r = −0.25, *p* < 0.001).

## 4. Discussion

The female sixth-grade students examined in the CReActivity study showed good levels of PA with 90.4% fulfilling the WHO recommendation on a weekday and 57.4% on a weekend day. Healthier PA and SB were significantly related to lower BMI. PA behavior was better than what had been assumed based on previous studies in Germany [15,16,17]. MVPA values of the present study were also higher compared to the international HELENA study, which reported a daily average of around 50 min of MVPA for adolescent girls from nine European countries including Germany [42]. Accelerometer-measured PA of participants aged 9 to 13 years from the ENERGY project conducted in five European countries indicated clearly lower mean MVPA values of 36 and 25 min on weekdays and weekend days, respectively [43]. However, the MVPA values of the present study were highly similar to those of an international systematic review by Brooke, Corder, Atkin and van Sluijs examining accelerometer-measured PA of school-aged children [21]. In the 36 studies included in their meta-analysis, most of them conducted in Europe and North America, subjects exhibited on average 82.3 min of MVPA on weekdays and 68.3 min of MVPA on weekend days. However, the standard deviation of the means amounts to 44 min, indicating a substantial variance in the MVPA values found by the different studies. The high ST values of more than eight hours in our study fit the results of a study with data from 40 countries in Europe and North America by Hallal and colleagues [44]. They estimated that two thirds of adolescents aged 13 to 15 already gathered at least two hours per day of ST only by watching television. With an average of around nine hours of daily ST accounting for 70% of girls’ total wear time, the results of the HELENA study were highly similar to the results presented in our study [42]. The substantial amount of ST in the present study was even exceeded by that reported in the results from female adolescents included in the Spanish UP&DOWN study, who exhibited more than seven hours of leisure-time SB averaged over weekdays and weekend days [45].

Even though the average PA level during the weekend was above the WHO recommendation in the present study, there was a significant decrease of LPA, MVPA and ST and WT compared to weekdays. The direction and magnitude of the difference in MVPA levels between weekdays and weekend days are in line with the results of the systematic review by Brooke et al. [21], in which MVPA was on average 14 min higher on weekdays than on weekend days. The pooled standardized mean difference of d = 0.42 reflects a small-to-medium effect which is also comparable to the one in our study. Thus, the assumed decline in PA from weekdays to weekend days can be confirmed.

However, the qualitative intensity of PA while being awake and active was only slightly lower on a weekend. Instead, the length of time in which the female adolescents engaged in PA was shorter. The participants of our study exhibited on average about 19 min less MVPA on a weekend day compared to a weekday. Similar to previous studies [21], this difference can mainly be traced back to the commute to and from school. Between 7 a.m. and 8 a.m., adolescents gathered around nine minutes more MVPA on schooldays compared to weekend days. In line with this, WT decreased by two hours from weekdays to weekends because of longer sleep duration in the morning [46]. It seems that students got up on a weekend day at the time they arrive at school on a weekday. This was shown on the curves of the respective activity patterns concerning MVPA and VMCPM, which approximate each other towards 8 a.m. (Figure 2 and Figure 3). As soon as they were awake, the PA pattern on a weekend day was comparable to that on a school day, with the exception of the period between 1 and 2 p.m., which can probably be attributed to the break between morning and afternoon classes or the way back home from school. During this hour, the girls exhibited almost four minutes more MVPA on school days. The remaining difference in MVPA of around six minutes between weekdays and weekend days was spread out during the course of the day. When PA behavior during time segments with different durations is compared, it is recommended to consult a relative measure like VMCPM next to an absolute measure like MVPA [21]. Since the participants averaged two hours more WT on weekdays than on weekend days, the respective difference in VMCPM was examined. Although exhibiting a statistically significant difference in favor of weekdays, the underlying effect was small and without practical meaning as the difference amounted to 20 counts per minute. This result is again supported by the systematic review by Brooke et al., who found a mean difference of about 30 CPM in favor of weekdays [21]. To put this into perspective, the range for LPA in the cut points from Hänggi et al. extends from 180 to 3360 VMCPM, which shows that the detected difference in VMCPM between weekdays and weekend days was, although statistically significant, practically negligible [10]. Additionally, although on weekdays the portion of MVPA during WT was significantly higher and the portion of ST during WT was significantly lower, these differences were small and did not indicate a qualitatively different dimension of PA on weekdays. For example, Figure 2 and Figure 3 illustrate that the intensity of PA which students exhibited between 7 and 8 a.m. on weekdays did not substantially exceed the intensity levels they showed during an average weekend afternoon.

The MVPA level in the examined sample indicates a healthy PA behavior. However, whereas the WHO recommendation [1] regarding PA was on average fulfilled, the German recommendation of 90 min daily MVPA [2] was only reached by a minority of the present sample. Moreover, PA behavior on weekend days was clearly worse than on weekdays. Since the WHO recommendation is understood as a minimum value and because additional health benefits can be expected with higher PA levels [1], the need for sustained PA promotion persists. Additionally, in view of the fact that SB partially implies its own risk factors [7], further drawbacks become apparent. The vast majority of the participants failed to meet the recommendation regarding ST during recreational time [2]. Taking into account that the recommended limit of two hours per day especially refers to avoidable ST, like motorized transport or screen time, the girls tended to approximate this recommendation on weekdays rather than on weekend days. On average, the students gathered around three hours of ST between 4 and 9 p.m. on a weekday. In the UP&DOWN study, educational-based SB (doing homework, studying and reading) accounted for one third of the girls’ leisure-time SB [45]. Thus, by applying these proportions to the SB results presented here, a mentionable proportion of the girls came close to or even fulfilled the ST recommendation when the approximated time spent on having dinner and other necessary activities, like doing homework, was subtracted from their average after-school ST. The average ST on a weekend day of more than seven hours, however, exceeded the recommendation by far. Thus, the assumption that the students did not fulfill the German recommendation regarding ST was largely confirmed, especially for weekend days.

The results indicate that the girls did not spend available time on weekends or after school specifically on being physically active. The increase in MVPA after school compared to school hours was small. Similarly, ST after school was only slightly lower than during school hours. MVPA on a weekend afternoon was nearly the same as during after-school hours. ST on an average weekday afternoon was on the same level as on a weekend day. These findings resemble the results of the HELENA study and two studies conducted in Great Britain with samples comprising similar age groups. In the PEACH study, the portion of MVPA after school was slightly higher than during school hours and during a weekend afternoon. In female participants of the HELENA study and in the PEACH study, the percentage of ST was only slightly lower after school than during school hours. Furthermore, in these samples, as well as for the girls of the SPEEDY study, the portion of ST during after-school hours was on the same level as on a weekend [47,48,49]. In short, weekday MVPA in the afternoon was not much higher than during school hours, ST did not significantly decrease in the afternoon. A comparison of afternoon PA and SB between weekdays and weekend days exhibited negligible differences. The highly comparable results of the systematic review of Brooke et al. [21] regarding the in-school versus out-of-school comparison and the out-of-school versus weekend comparison led the authors to the similar conclusion that children did not choose to spend their additional free time during weekends on PA. An alternative explanation would be that leisure time hardly exists for students since they spend too much time doing homework in the afternoon and during the weekend [50].

However, even if hardly any free time is available for students, there should still be several ways that PA can be promoted. Given that, during school hours, students’ PA was low and ST was high, the school represents one setting where interventions, especially of a multicomponent character, can be effective. One possible component is the promotion of active transport, which has led to an increase in PA in a variety of studies [25] and is also in line with the PA peaks on weekdays in the present sample. Another intervention component promoting PA directly within school hours could be classroom-based interventions implemented in the regular curriculum [26]. Further components of interventions in the school setting could include activity breaks, after-school programs, changes in the school environment and improvements in the quality of PE lessons [25]. One possibility to positively affect PE quality is to conduct PE lessons based on self-determination theory (SDT; [51]). SDT is one of the theoretical frameworks that is commonly used in the educational context [52]. Applied to PE, it may initiate a chain of effects resulting in higher PA levels. SDT suggests that if the three so-called basic psychological needs (BPNs), autonomy, competence and relatedness, are satisfied, autonomous motivation is higher, which finally promotes the respective behavior. Studies in a PE context have shown that PE teachers can satisfy students’ BPNs by applying a need-supportive teaching behavior [53]. This way, students’ autonomous motivation during PE can be increased [53,54]. Subsequently, autonomous motivation for PA in leisure time can be enhanced as well, which finally results in higher PA levels after school and on weekends [53,54,55]. In this way, PE can enable a transfer of PA-enhancing effects during school hours into leisure time, and thus the family setting. In this setting, further contributions to PA increase and SB decrease can be made by appealing to the role of the parents. It has been shown that parental support can increase PA and reduce SB of children [25,56]. Additionally, the PA and screen time of children are positively related to the PA and screen time of their parents [25]. These findings can help build the foundation for interventions in the family setting in order to find appropriate responses to high levels of SB during weekends, which were also found in the present study.

Accelerometer-based PA data assessment is more objective than a self-report approach [9]. However, device-based measurement techniques are not necessarily free of possible biases. One example would be a possible motivational effect of the PA measurement itself, referred to as the mere-measurement effect [57]. The students knew that the accelerometers measured PA and they were also aware of the fact that higher PA levels are socially rewarded. Social desirability and reactivity leading to behavior adaption in the sense of a “Hawthorne effect” may have then led to students trying their best to be physically active [58,59]. This effect might have occurred despite the fact that the students who were enrolled in the intervention study were not told whether they formed part of the intervention or control group, as an increase in PA due to its mere measurement can happen in members of both groups [60].

Additionally, and even more importantly, researchers make many data processing and analysis decisions when creating accelerometer-derived estimates of ST and PA [30,61]. These include decisions concerning the sampling rate, the choice of epoch length, the selection of an algorithm to differentiate accelerometer WT from non-WT and the cut-point definitions to determine the amount of time spent in ST and PA intensity levels. The choice of the methodological procedure was an important factor for explaining the high MVPA values found in this study in comparison to previous studies in Germany and Europe.

Using short epoch lengths leads to a higher resolution of the measurement and therefore increased MVPA values [61]. Analyzing accelerometer data of 7 to 11-year-olds, Banda et al. could show, with the example of the often-applied Evenson et al. cut points [62], that the average MVPA value using a 60-s epoch length was only 67.2% of the value using 1-s epochs. Using other cut points the difference was even more severe (Romanzini [63]: 65.8%; Treuth [64]: 50.1%; Puyau [65]: 47.4%; Mattocks [66]: 39.6%) [61].

Furthermore, even when activity cut points are applied together with the epoch length originally used to validate the respective cut points, MVPA values still differ to a great extent between the cut points. Banda and colleagues showed in their study that when using Romanzini cut points, 107.6 min of MVPA per day were calculated in comparison to between 15.2 and 59.9 of MVPA minutes per day when using the other four activity cut points [61]. However, the obvious effect of different minimum amounts of counts per minute that have to be reached to classify activities as being at least moderate cannot fully explain the differences in the calculated MVPA values. This is when the question of using vertical axis counts or VM counts comes into play. Romanzini et al. offered cut points which were validated using VM counts, whereas the other four cut points only used the counts of the vertical axis. When analyzing PA behavior by VM cut points, ST values are lower and MVPA values are higher compared to using vertical axis counts only [67]. Using VM scores, the precision of the measurement is enhanced as movements in everyday life are more frequently performed in an anterior–posterior and medio-lateral direction than in a vertical direction. Likely because of that, the MVPA estimates of elderly women examined by Keadle et al. using VM cut points were twice as high as when considering vertical axis counts only. This is supported by a study of Migueles et al. showing that the VM activity cut points from Romanzini et al. and Hänggi et al. lead to the highest MVPA estimates compared to other established cut points [68]. These findings lead to the conclusion that PA estimates using VM cut points and vertical axis cut points cannot be directly compared [67].

As the main strength of our study, we followed recent recommendations calling for high resolution and precision in PA data assessment and analysis by sampling with 30 Hz, using a 1-s epoch length and applying the VM activity cut points from Hänggi [30,61]. In contrast, the German GINIplus and IDEFICS studies used 60-s epochs and vertical axis cut points [15,16]. The German KiGGS study applied VM cut points but only a 15-s epoch length [17,69]. Both the European HELENA study (including data from adolescents of one German city) and the ENERGY project only measured vertical movements and analyzed accelerometer data with an epoch length of 15 s [42,43]. The Spanish UP&DOWN study used both triaxial and uniaxial accelerometers and a 10-s epoch length [45]. The different decisions regarding accelerometer data sampling and processing criteria have undoubtedly contributed to less accurate and thus lower activity estimates in previous German and European studies compared to our results. The data from the HELENA study have now been examined again with different cut-point definitions, including different epoch lengths [70]. Based on data from the exact same sample, fulfillment of the WHO recommendation for MVPA ranged between 5.9% and 37%, according to the applied cut points. In view of these findings, the authors questioned the results of the HELENA study by suggesting that the reported PA levels could have been significantly different if they had used other epoch lengths, which, however, is only one of several processing criteria in accelerometer data analysis. The authors further concluded that inaccuracies in accelerometer analysis could lead to wrong inferences when examining the relationship of PA levels and health outcomes and thereby underlined the importance of precise accelerometer analyses [70].

With regard to SB, the results of our study and those of previous studies do not differ as much as with regard to PA. The German GINIplus and IDEFICS studies indicated a comparable SB [15,16]. ST values of our study are similar to those of the HELENA study [42]. Participants of the ENERGY project exhibited highly similar ST values both on weekdays and on weekend days. Female participants of the Spanish UP&DOWN study exhibited higher SB levels, with more than seven hours of ST during leisure time only [45]. The reason for the higher agreement between studies regarding ST values compared to PA values is that estimation of ST values is less affected by different accelerometer sampling and processing criteria. For example, if a person is sitting still for an uninterrupted period of time, this is considered as ST regardless of the chosen sampling rate, epoch length or activity cut points as there is no activity at all. Different choices in these criteria do not have an effect until the person shows short interruptions of ST by intermittent periods of PA, which may only then be identified as activity if the chosen sampling and analysis criteria allow for it. An intermittent PA behavior is more common among children and adolescents, which underlines the necessity of accelerometer analyses with a high resolution for these populations especially [30].

The main limitation of this study is that the results only refer to female sixth graders and cannot be considered as being representative for the German population of children and adolescents. Analyses with samples including an equal number of female and male participants, as well as participants from different age groups, regions and school types, are necessary to allow for nationwide and precise insights into PA and SB of children and adolescents in Germany. Another limitation of the study is the lack of a log recording the times when the accelerometer was worn or a diary depicting the course of the day of every student. This way, the interpretation of the differences between PA and SB on weekdays and weekend days could have been corroborated by the students’ recordings. Furthermore, since data were collected during autumn and winter, data of this study do not reflect the possible seasonal variations in PA and SB. PA of children and adolescents tends to be higher in spring and summer compared to autumn and winter [71,72]. SB normally reaches its highest level during autumn and winter, with seasonal differences being greater on weekends [72]. Whereas this might actually attenuate or even balance the mere-measurement effect discussed above [57], it can certainly be considered as a limitation in terms of the representativeness of the PA and SB data regarding seasonal variations.

## 5. Conclusions

The results of this study lead to three main conclusions. First of all, PA behavior of the female sixth graders participating in the CReActivity project was better than expected in light of previous German and international studies. The vast majority fulfilled the WHO guideline on PA on weekdays and more than half of the students were sufficiently active on weekend days. Secondly, SB of the participants was worrying, especially during school hours and on weekend days. Considering that the detrimental effects of high SB on health are partially independent of the effects of physical inactivity, multicomponent interventions need to find solutions that enable students to reduce their SB levels at school and engage in less sedentary activities with family and friends during weekends. Third, the accelerometer assessment of PA and SB would have led to less precise results if the sampling and processing criteria of previous German and international studies had been applied. This questions the accuracy of previous findings and highlights the necessity of larger representative studies that follow the most recent recommendations regarding accelerometer-based measurement of PA and SB. Only highly precise measurements of PA and SB levels and patterns allow for an accurate problem analysis that can subsequently serve as foundation for creating suitable and effective solutions.

## Figures and Tables

**Figure 1 ijerph-18-00032-f001:**
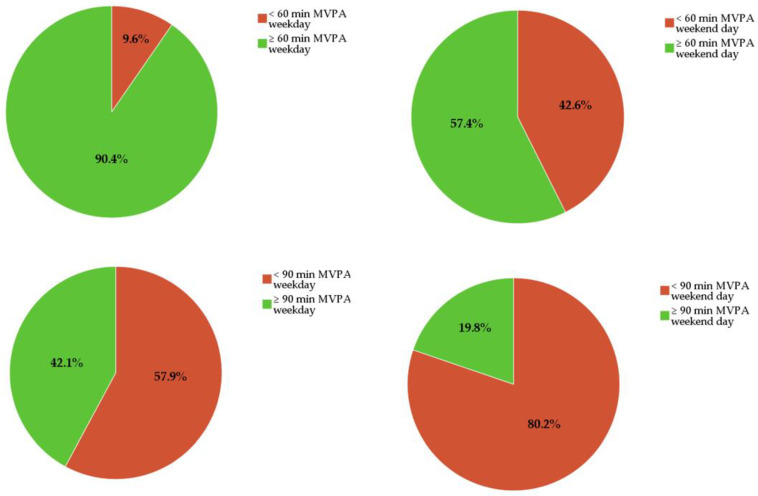
Percentages of the sample fulfilling the respective PA recommendations. Green = recommendation fulfilled, red = recommendation not fulfilled; MVPA = moderate-to-vigorous physical activity.

**Figure 2 ijerph-18-00032-f002:**
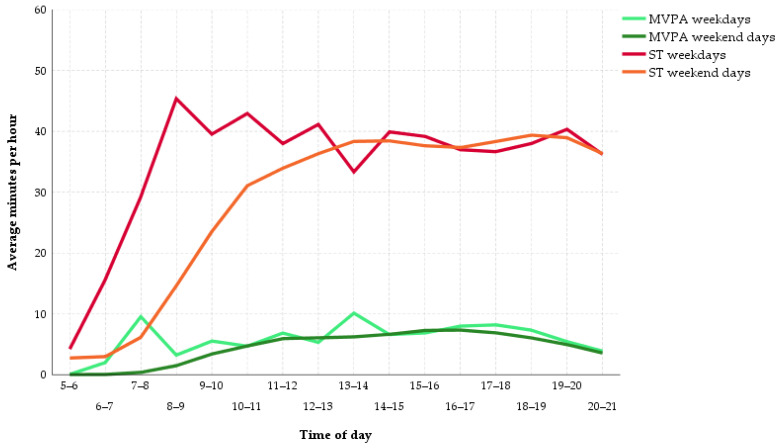
Sedentary behavior (SB) and moderate-to-vigorous physical activity (MVPA) patterns on weekdays and weekend days. ST = sedentary time.

**Figure 3 ijerph-18-00032-f003:**
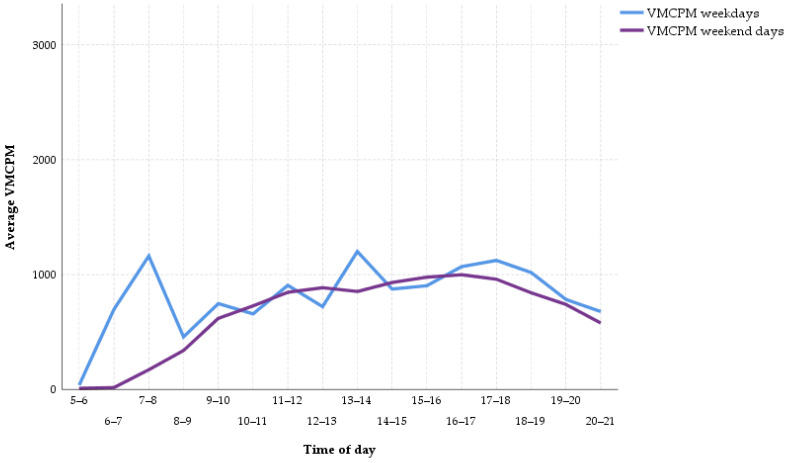
Vector magnitude counts per minute (VMCPM) pattern on weekdays and weekend days.

**Table 1 ijerph-18-00032-t001:** Sedentary and physical activity behavior in minutes and in percent of wear time (mean (SD)) on average weekdays, weekend days and in total (N = 425).

	Weekday	Weekend Day	Wilcoxon Test Z-Value, Effect Size r	Total
ST/day (min)	515.0 (72.5)	434.8 (91.6)	−14.07 ***, 0.48	493.7 (67.6)
LPA/day (min)	177.0 (50.4)	147.0 (51.6)	−13.10 ***, 0.45	168.7 (47.2)
MVPA/day (min)	87.5 (23.2)	68.4 (30.2)	−13.20 ***, 0.45	82.3 (22.4)
VMCPM (min)	875.2 (225.3)	855.2 (323.3)	−2.72 **, 0.09	867.9 (221.9)
WT/day (min)	779.4 (57.5)	650.1 (94.7)	−16.21 ***, 0.56	744.6 (50.7)
ST/WT (%)	66.0 (7.7)	66.9 (9.9)	−2.56 *, 0.09	66.3 (7.6)
LPA/WT (%)	22.7 (6.4)	22.6 (7.4)	−0.63, 0.02	22.7 (6.3)
MVPA/WT (%)	11.2 (2.9)	10.5 (4.4)	−5.39 ***, 0.18	11.1 (2.9)

Note. SD = standard deviation; min = minutes; ST = sedentary time; LPA = light physical activity; MVPA = moderate-to-vigorous physical activity; VMCPM = vector magnitude counts per min; WT = wear time on valid days; */**/*** = *p*-values for difference between weekday and weekend day <0.05/<0.01/<0.001.

## Data Availability

The data presented in this study are available online as Appendix A.

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
