# Peer review of "Accelerometer-Measured Physical Activity and Sedentary Behavior Levels and Patterns in Female Sixth Graders: The CReActivity Project"

_ijerph, 2020, doi:10.3390/ijerph18010032_

Round 1

Reviewer 1 Report

The abstract is organized in a well-structured format. However, taking into consideration the objective of the study, it is suggested that this section may also include a general comment about the results of sedentary behavior to establish a more complete understanding of the evidenced results.

The section "Introduction" presents a sufficient contextualization of the research problem, referring to previously published studies to broaden the context and provide an empirical basis for the subsequent development of hypotheses. In this sense, even if the present study is of descriptive design, the elaboration of the hypothesis is recommended. Also, readers of a research report are usually concerned with the relevance of the problem, both in theory and practice. In this sense, it is suggested that the presentation of the relevance of the main subject understudy to the current practical knowledge, concerning interventions with young people, can also be supported by other previously published evidence, giving it better robustness. The different operational definitions of sedentary time that the studies use the result in different estimates of that time, making it difficult to compare the studies and, consequently, the progression in this field of investigation. Besides, as research on sedentary behavior has grown, other related terms have begun to appear, such as screen time, sedentary behavior pattern, and interruption of sedentary behavior. This set of concerns has resulted in the need to standardize the terminology to be used in the study of sedentary behavior. Thus, it is important to understand that sedentary behavior is distinct from the concepts of physical inactivity and physical activity and that the health consequences resulting from sedentary behavior are distinct from the lack of physical activity. In this sense, to have a differentiation of related concepts, it is suggested to clearly distinguish the concepts of physical inactivity from sedentary behavior. Furthermore, it is recommended that the definition of sedentary behavior be updated according to the Sedentary Behavior Research Network Terminology Consensus Project. Tremblay, M., Aubert, S., Barnes, J., Saunders, T., Carson, V., Latimer-Cheung, A., ... SBRM Terminology Consensus Project Participants. (2017). Sedentary Behavior Research Network (SBRN) - Terminology Consensus Project process and outcome. 49 International Journal of Behavioral Nutrition and Physical Activity, 14: 74.

The "Participants" section allows other researchers to select a virtually identical sample if they choose to repeat this study, under the same conditions. However, it is still necessary to clarify some issues. It would be important to clarify which sampling procedure is used. Were the participants randomly or intentionally selected? Since sample size plays an important role in the ability to make accurate inferences, has any statistical procedure been performed to calculate the required sample size? There is a marked discrepancy between the number of male and female participants. Also, most of the data from female participants came from an intervention study aimed at encouraging physical activity. How do these aspects contribute to the fact that the results shown represent the reality, in terms of levels of physical activity and sedentary behavior, of other young Germans of the same age?

The presentation of the results in the textual form is carried out appropriately and perceptibly for the reader. Also, the figures and tables show the essential data, not revealing duplicate data in the tables and text. However, data were collected about the BMI and the socioeconomic status of the parents of the youth. The analysis of these data could also be considered to describe with another level of detail the results presented. The "Discussion" section relates the results of this research to previously published studies related to the subject of study. However, even though there are methodological differences between the present study and the studies described in the literature related to the subject of the present study, it is suggested that the potential significance of the practical application of the results obtained should be significantly deepened. The authors report that young people comply with the WHO recommendations regarding the levels of moderate-intensity physical activity in force on weekdays compared to weekend days. Some explanations for these results are given. However, to develop multilevel approaches that allow more comprehensive and effective interventions to reduce sedentary behavior of young people and that encourage increased levels of physical activity, it would be interesting to provide a more in-depth theoretical/empirical explanation, taking into account, for example, ecological and psychosocial models explaining sedentary behavior and physical activity practice. How did the authors manage to specify the sedentary activities that the participants performed? It is suggested to clarify (line 307: Taking into account that the recommended limit of two hours per day especially refers to avoidable ST like motorized transport or screen time, the students tend to approximate this recommendation rather on weekdays than on weekend days.) (line 311: Subtracting the time spent on having dinner and other necessary activities like doing homework, a mentionable proportion of students comes close to or even fulfill the recommendation.). The ecological models explain that among the different determinants proximal to the individual and the more distal factors that influence the practice of physical activity, the climatic conditions can influence the level of physical activity. In this sense, considering that the results were collected during the fall and winter may constitute a limitation of the results presented. The conclusion of the study is not perceptible. It is recommended that, instead of the "Conclusions" section presenting methodological considerations, becoming redundant concerning what was discussed in the "Discussion" section, an interpretation of the results evidenced here be carried out.

Reviewer 2 Report

Dear authors, congratulations on your work. The high percentages obtained in their study of PA were the reason for wanting to know how they carried out their study. In a general way, in my opinion, there is too much information, taking into account that, as you say, no comparison can be made with other studies. They should consider that since the selection of cut-off points is well discussed, it may be that the paper has to be redirected, then justifying the need to unify criteria to establish cut-off points, rather than to expose a certain AP situation that cannot finally be compared. and that is undoubtedly striking. Below, I suggest a series of changes for a better understanding of their work.

Summary

Please note that the abstract is the cover letter for your work.

Write the objective properly. Since 4 objectives are reflected in the manuscript, put the general objective appropriately for example.

Please clarify which was the sample on which the study was carried out.

Take the same criteria to write numbers and percentages (in letter or in number)

If possible, enrich the results presented in the summary and provide statistical data.

Introduction

Regarding line 50-51, I think I understand that it refers to the different cut-off points that can be assessed to define the quantity and quality of physical activity. If so, reflect it in a simpler way for the reader.

Line 60-61: The self-reported data usually reflects more physical activity than actually performed (measured with a device). This makes the results obtained in this study really striking. Indeed, the data from the KiGGS study and the HBSC survey justify the efforts to promote physical activity in this population group. With the data obtained in this paper, an intervention to promote physical activity is not justified, since 90% of the population meets the WHO recommendations measured with a device.

On line 61, review the abbreviations.

Also could you specify the age ranges of the GINI and IDEFICS studies? The reason for requesting it is that, as reflected in the comparison between the KiGGS and the HBSC in adolescents, there is a lower level of physical activity.

Line 65-66: I understand then that the KiGGS study also performed accelerometer measurements. This paragraph needs to be clarified and organized by selecting the information they want to provide and not providing percentages that may later confuse the reader. For example, if you want to emphasize studies with a device, do so but try not to saturate with other data. I also insist on clarifying the age ranges. The activity of a 5-6 year old child is not the same as that of a 14-15 year old boy. Try to center it.

Material and methods

Participants: What type of sampling did you do? Justify the low percentage of children.

In my opinion, it is best to provide the participation rate after obtaining the final sample.

Line 99: I understand then that only education centers in the periphery were evaluated, not within the big cities

Be more specific in providing the ethical data, do not intersperse it with the explanation of the study sample

The BMI data and the NSE data are not relevant in this paper. Assess the need to include them.

Measures: Line 125 should go to Procedure. On line 145, refer again to ethical permissions when you have already mentioned it in the participants section. Try to unify this information. On line 179, it refers again to how the final sample was configured, when it had already been done in the Participants section. Unify information

The results are well explained

The discussion is correct, it is well defended with bibliography, but it is very strange to think that 90% of German adolescents do the recommended PA. You justify it with the trips to school. My question is, what criteria did you use to consider MVPA? I understand that they move by walking (at most at a brisk pace). Could it be that there is a methodological error in the MVPA cut-off points?

Reviewer 3 Report

I appreciate the opportunity to review this interesting paper entitled "Objective Levels and Patterns of Physical Activity and Sedentary Time in Sixth Graders in Southern Germany". In this work the authors explored the accelerometer-measured levels and patterns of physical activity and sedentary time of sixth graders in Southern Germany. Clearly a lot of work went into its construction. However, there are limitations that have to be taken into account. Below are my major comments for the authors.

  1. The main limitation of this study is its questionable novelty. There are a lot of cross-sectional studies which report device-measured physical activity levels in European children and adolescents (e.g., papers derived from HELENA study or UP&DOWN study). Likewise, several of them show patterns of physical activity and sedentary time in children and adolescents from European countries (i.e. weekend, weekdays, school time, afterschool time…).
  2.  The sample supposes another important limitation. It is not statistically representative from any region. Although the study includes device-measures of physical activity, the sample size is not very large in comparison with similar cross-sectional and longitudinal studies. Moreover, only 77 of the 622 participants are boys! Because gender influences physical activity levels, the sample characteristics could bias the results. I suggest remove the boys or to differentiate the results by gender.
  3. The number of references is too large for an original paper. I recommend substantially narrowing down the reference list, removing old or expendable citations and incorporating specific, recently published references.
  4. Moreover, authors should revise other minor issues of their manuscript (e.g., reference format in reference list; to avoid references in the conclusion section; aim of the study at the end of the introduction section; to improve the quality of the figures; to replace “objective” by “accelerometer-measure”; include the accelerometer processing criteria in the 2.2.1 section instead of in the data analysis; to explain why sixth graders…).

The use of accelerometers to measure physical activity levels is the main strength of this manuscript. Below my comments on accelerometry to the authors:

  • Although the use of accelerometers is the main strength of this manuscript, strictly speaking it is not an “objective” measure because there are subjective processing criteria of accelerometer data.
  • There are recent studies with accelerometers with objectives very similar to those of this manuscript (e.g. from the UP&DOWN study). They should be used to discuss the results.
  • Information about accelerometer data collection and processing criteria should be in the measures section and not in the data analysis section.
  • The selected processing criteria are valid, but why did you select “at least eight hours of wear time (WT) being required for a valid day” instead of a more ambitious amount of time? Eight hours are only 1/3 of the day so the data loss seems too great for a descriptive study.

Best regards.

Round 2

Reviewer 1 Report

The authors have made most of the changes requested in the previous review report. Therefore, the following comments are added:

Regarding the definition of sedentary behavior, it would be useful that the authors could complement it with the concept of "Sedentary behavior pattern" by Tremblay et al. (2017), against the title of this research report. Also, it is suggested that the object of study can be contextualized in the introductory section, taking into consideration the ecological models that explain sedentary behavior and the practice of physical activity.

It is still unclear how the authors, using the accelerometer, were able to specify the sedentary activities that the participants performed. (By approximation, a mentionable proportion of students comes close to or even fulfills the ST recommendation, when the time spent on having dinner and other necessary activities like doing homework is subtracted from their average after-school ST.).

There is a high imbalance between the number of female participants and male participants. The authors report that this is a factor that can lead to a bias in the results, concerning physical activity and sedentary behavior, and cite two published studies (18,26). It would also be interesting to report on the trend of the results already published regarding gender influence on these variables.

The way the conclusion of this report is described is still not perceptible. It is suggested that a general interpretation of the results evidenced here be carried out, without including references of previously published literature, leaving them for the "Discussion" section.

Reviewer 2 Report

Congratulations on your work.

My doubts have been satisfactorily resolved. They have correctly defended their results with the appropriate scientific literature.

Author Response

Thank you for your positive evaluation.

Reviewer 3 Report

Dear Editor,

I thank the authors for providing good answers to my comments and in general making appropriate changes to their manuscript. I have some few additional comments:

  1. I suggest removed the title of the project in the manuscript title. Another option could be to include it after a colon.
  2. In this new version of the manuscript, there continues to be a problem with the number and specificity of references. The maximum number of references that journals recommend for an original paper are 30 or 40 references. Although IJERPH have not a limit of references, the manuscript includes 81 references! I recommend a thorough review of the need and specificity of each and every one of them.
  3. In despite of the previous comment, there are relevant references about PA or SB patterns omitted in the manuscript (e.g., https://doi.org/10.1080/02640414.2020.1734310; https://ijbnpa.biomedcentral.com/articles/10.1186/s12966-015-0204-6).

Best regards.
